# State of the Art and New Trends from the Second International StemNet Meeting

**DOI:** 10.3390/ijms25042221

**Published:** 2024-02-13

**Authors:** Ivana Ferrero, Filippo Piccinini, Pasquale Marrazzo, Manuela Monti, Caterina Pipino, Alessia Santa Giovanna Banche Niclot, Camilla Francesca Proto, Enrico Ragni, Ralf Hass, Giulia Maria Stella, Priscilla Berni, Ana Ivanovska, Katia Mareschi

**Affiliations:** 1Stem Cell Transplantation and Cellular Therapy Laboratory, Paediatric Onco-Haematology Division, Regina Margherita Childrens’ Hospital, City of Health and Science of Turin, 10126 Turin, Italy; iferrero@cittadellasalute.to.it (I.F.); cproto@cittadellasalute.to.it (C.F.P.); 2IRCCS Istituto Romagnolo per lo Studio dei Tumori (IRST) ‘‘Dino Amadori”, 47014 Meldola, Italy; filippo.piccinini@irst.emr.it (F.P.); manuela.monti@irst.emr.it (M.M.); 3Department of Medical and Surgical Sciences (DIMEC), University of Bologna, 40126 Bologna, Italy; pasquale.marrazzo2@unibo.it; 4StemTeCh Group, Center for Advanced Studies and Technology-CAST, Department of Medical, Oral and Biotechnological Sciences, University G. D’Annunzio Chieti-Pescara, 66100 Chieti, Italy; caterina.pipino@unich.it; 5Department of Public Health and Paediatrics, University of Turin, 10126 Turin, Italy; alessiagiovannasanta.bancheniclot@unito.it; 6Laboratorio di Biotecnologie Applicate all’Ortopedia, IRCCS Istituto Ortopedico Galeazzi, Via Cristina Belgioioso 173, 20157 Milano, Italy; enrico.ragni@grupposandonato.it; 7Biochemistry and Tumour Biology Laboratory, Department of Obstetrics and Gynecology, Hannover Medical School, D-30625 Hannover, Germany; hass.ralf@mh-hannover.de; 8Department of Internal Medicine and Medical Therapeutics, University of Pavia Medical School, 27000 Pavia, Italy; giuliamaria.stella@unipv.it; 9Unit of Respiratory Diseases, Cardiothoracic and Vascular Department, IRCCS San Matteo Hospital Foundation, 27100 Pavia, Italy; 10Department of Veterinary Sciences, University of Parma, 43121 Parma, Italy; priscilla.berni@unipr.it; 11Regenerative Medicine Institute (REMEDI), University of Galway, H91 TK33 Galway, Ireland; ana.ivanovska@universityofgalway.ie

**Keywords:** stem cells, annual meeting, conference report, Forum of Italian Researchers on Mesenchymal and Stromal Stem Cells (FIRST), Italian Mesenchymal Stem Cell Group (GISM), International Placenta Stem Cell Society (IPLASS), Stem Cell Research Italy (SCRI)

## Abstract

The Second International StemNet (Federation of Stem Cell Research Associations) meeting took place on 18–20 October 2023 in Brescia (Italy), with the support of the University of Brescia and the Zooprophylactic Institute of Lombardy and Emilia Romagna. The program of the meeting was articulated in nine sections: (1) Biomedical Communication in Italy: Critical Aspects; (2) StemNet Next Generation Session; (3) Cell-Free Therapies; (4) Tips and Tricks of Research Valorisation; (5) Stem Cells and Cancer; (6) Stem Cells in Veterinary Applications; (7) Stem Cells in Clinical Applications; (8) Organoids and 3D Systems; (9) induced pluripotent stem cells (iPCS) and Gene Therapy. National and International speakers presented their scientific works, inspiring debates and discussions among the attendees. The participation in the meeting was high, especially because of the young researchers who animated all the sessions and the rich poster session.

## 1. Introduction

The Second International StemNet meeting took place on 18–20 October 2023 in Brescia (Italy) in a three-day conference full of scientific novelty and active participation, with the support of the University of Brescia and Zooprophylactic Institute of Lombardy and Emilia Romagna. StemNet is a federation of the four main associations of stem cell research in Italy (Forum of Italian Researchers on Mesenchymal and Stromal Stem Cells (FIRST), Italian Mesenchymal Stem Cell Group (GISM), International Placenta Stem Cell Society (IPLASS), and Stem Cell Research Italy (SCRI)), which share and synergise experiences to enhance both the quality and impact of research in this advancing field. The meeting reflected the constructive relationship among the four associations and aimed to improve the exchange of relevant and up-to-date information in the field of advanced cell therapy. In particular, the program included nine sessions focusing on basic and translational stem cell research in human and veterinary medicine: “Biomedical Communication in Italy: Critical Aspects”; “StemNet Next Generation Session”; “Cell-Free therapies”; “Tips and Tricks of Research Valorisation”; “Stem Cell and Cancer”; “Stem Cells in Veterinary Applications”; “Stem Cells in Clinical Applications”; “Organoids and 3D Systems”; “iPSC and Gene Therapy”. The meeting included discussions on the critical aspects of biomedical communication and research valorisation, and a special “next generation” session organised and intended for young scientists. National and International experts and renowned speakers shared their works, inspiring debates and discussions. In total, the participants were more than 150, including attendants and attendees, of which more than 40 were young researchers under 35 years, shedding further light on the importance and interest created by this congress. 

## 2. Session Summary

The meeting was articulated in nine different sessions in order to diversify the various study and research sectors (clinical, cell culture and extracellular vesicles, veterinary, oncology, etc.) and to broaden the discussions and experiences of clinicians and researchers in the specific fields of study.

The congress was introduced by Prof. Augusto Pessina, president of StemNet, who expressed gratitude to all the participants and introduced the first session underlying the importance of scientific communication in our country and how it has to be led by experts.

### 2.1. Session 1: “Biomedical Communication in Italy: Critical Aspects”

The first speaker was Dr. Daniele Banfi, a scientific journalist for Fondazione Veronesi and Gedi group newspapers, who talked about how often the journalists who write about scientific topics and current health information do not have a scientific background, aiming to prepare clickbait titles and content which frequently increase the spread of fake news. It is important to avoid misinformation in the general public audience as this causes panic and disbelief (O-01).

This speech was followed by Dr. Agnese Collino, who is a communicator on social media platforms. She described the importance of creating easy-to-understand content that can be more appealing to the general public. Researchers can also use these platforms to spread the knowledge of new discoveries (O-02).

The last speaker was Dr. Alessandro Iapino, the head of the “Ospedale Bambino Gesù” press office. He gave his point of view as the head manager of all the communication channels that disclose information from this institution. He explained the challenges not only for the quality of the published information but also for those who want to be the author of such topics (O-03). 

### 2.2. Session 2: “Stemnet Next Generation Session”

Young StemNet members organised this section to provide a platform for early career researchers to showcase their published articles to the StemNet community. The speakers for this section have been selected through the “StemNet next generation contest 2023”. The young StemNet member committee evaluated the curriculum vitae of candidates under 40 years old who were the first/last/corresponding authors of submitted abstracts related to published articles. In addition to the speech of three contest winners, the committee invited an external outstanding speaker: Prof. Cecilia Laterza (University of Padua, Italy), a young Italian neuroscientist who recently won the European Research Council (ERC) Starting Grant.

The session began with Dr. Filippo Piccinini (IRST IRCCS “Dino Amadori”, Meldola, Italy, and University of Bologna, Italy), who chaired the session with Dr. Pasquale Marrazzo (University of Bologna). Dr. Piccinini described the rules of the “StemNet next generation contest 2023” established a few months ahead of StemNet. He summarised the main points as follows:-The candidates had to be researchers under 40 years (born not earlier than 1 January 1983);-They had to submit an abstract of an indexed scientific article in the field of stem cells;-Eligible abstracts had to be related to scientific articles published in the last three years (since 2020) (manuscripts just submitted or under review were not be eligible);-The candidate had to be the first author, co-first author, last author, co-last author, corresponding author, or co-corresponding author of the scientific article;-Each candidate could submit only one abstract but could be an author of other abstracts submitted by other researchers.

Piccinini concluded that the organisers also plan to repeat the contest experience in the next editions of the StemNet meeting.

Next, Dr. Marrazzo introduced the invited speaker, Prof. Cecilia Laterza from the University of Padova, Italy. She was chosen for her remarkable and illustrious career, with the belief that her achievements would serve as a wellspring of inspiration for aspiring young researchers, setting a shining example of what can be achieved in the field. Prof. Laterza is a young Italian scientist who has embraced an evolving career path, ranging from neurobiology to bioengineering. During her doctoral research, she explored induced pluripotent stem cell (iPSC)-based therapies and in vitro modelling of multiple sclerosis, utilising patients’ skin cells reprogrammed into brain cells. During her introduction at StemNet 2023, she gave an impressive presentation titled “FMR1 silencing and stability during human FXS brain development: from the early stage of pluripotency to 3D cortical organoids” [1]. Laterza summarised the main results of her current research projects and showed how her switch from biotechnology laboratories to an engineering laboratory helped her to better weigh possibilities in disease modelling and transplantation of stem cells as a therapeutic strategy for neurological diseases (O-04).

Dr. Marrazzo proceeded to introduce the three winners of the “StemNet next generation contest 2023”. These distinguished researchers were Dr. Vanessa Castelli, representing the University of South Florida College of Medicine (USA) and the University of L’Aquila (Italy); Dr. Caterina Pipino, hailing from the University Gabriele D’Annunzio Chieti-Pescara (Italy); and Dr. Giulio Di Minin, affiliated with the Swiss Federal Institute of Technology ETH Hönggerberg in Zurich, Switzerland. Dr. Castelli presented her article titled “Contraceptive drug Nestorone Enhances Stem Cell-Mediated Remodelling of the Stroke Brain by Suppressing Inflammation and Rescuing Mitochondria”. This work [2] was recently published in the journal *Free Radical Biology and Medicine* (O-05). Dr. Pipino discussed a recent publication titled “Effect of the Human Amniotic Membrane on the Umbilical Vein Endothelial Cells of Gestational Diabetic Mothers: New Insights on Inflammation and Angiogenesis” [3], which was published in *Frontiers in Bioengineering and Biotechnology* in 2022 (O-06). 

Lastly, Dr. Di Minin presented his research titled “TMED2 Binding Restricts SMO to the ER and Golgi Compartments” (O-07), published in *PLOS Biology*, 2022 [4].

The section came to an end by providing an invaluable platform for significant discussions about the future of the StemNet next generation section and the pivotal role of stem cells in disease modelling and future therapy. Notably, inquiries regarding the career trajectories of the speakers and their upcoming research ventures were abundant, igniting stimulating conversations within the vibrant StemNet community. Furthermore, this session served as a forum for the organisers to gather precious insights, setting the stage for future societal activities. It was unanimously agreed to move forward with the proposal for the “StemNet next generation contest”, solidifying the commitment to support emerging talents in the field.

### 2.3. Session 3: “Cell-Free Therapies”

The first session of the second day titled “Cell-Free Therapies” was focused on innovative cell-free approaches. Alternative approaches to living cells for specific applications represent an emerging field. These include the application of the released secretome, the extracellular vesicles (EVs), a combination of synthetic biology and material science as well as regenerative approaches able to reactivate existing progenitors and tissue-specific cells and reduce inflammation.

In this frame, Dr. Paolo Bergese (University of Brescia, Italy) gave a presentation on the potential of mesenchymal stromal cells (MSC)-derived extracellular vesicles (MSC-EVs) to reduce inflammation, promote healing, and improve organ function [5]. These features rely on the originating cell-tailored composition and structure of the EV membrane, which is composed of different lipids and proteins that are able to drive targeting and incorporating potential in recipient cells. To date, such sophisticated architecture is inaccessible to synthetic mimics, and the idea of using natural EV membranes as vehicle and target nanomaterials has been extensively studied. This innovative approach was described by Dr. Bergese, who introduced a synthetic-biogenic (hybrid) nanotechnology developed by his group based on dressing synthetic nanomaterials and nanodevices in a wetsuit made of MSC-EVs and their specific portfolio of molecules driving interaction and uptake in target cells (O-08).

The importance of crucial players in guiding cell activity was further described by Dr. Gabriele D’Uva (University of Bologna, Italy), who spoke about novel therapeutic strategies for cardiac regeneration through the dedifferentiation and proliferation of existing cardiomyocytes. This process resembles what is observed in lower vertebrates and prenatal/early postnatal mammals that are able to spontaneously regenerate injured hearts [6]. Dr. D’Uva and his group identified glucocorticoids as crucial players in the ability to regenerate cardiac tissues. Endogenous glucocorticoids activate the glucocorticoid receptor (GR), thereby promoting cardiomyocyte maturation and reducing proliferation. This clearly emerged in in vivo models where, through the inhibition of glucocorticoid activity either genetically by deleting GR or pharmacologically, cardiomyocyte proliferation was promoted, and scar formation after myocardial infarction was reduced. Thus, a new potential cell-free strategy for therapeutic interventions that can overcome limits in heart regeneration was proposed based on limiting GR activation by targeting this molecule (O-09).

Importantly, single factors are not the only crucial pillars for cell-free approaches when complex products, like the array of soluble cell-released factors or EVs, which are envisioned as therapeutics. In this perspective, Dr. Enrico Ragni (IRCCS Istituto Ortopedico Galeazzi, Italy) showed how MSC secretome and MSC-EVs embedded microRNAs may be modulated to enhance their therapeutic features for musculoskeletal disorders [7]. The secretome in recent years has emerged as a means to include originating MSC properties, offering the advantage of being a cell-free product. Several efforts have been made to empower the regenerative and anti-inflammatory potential. The work presented by Dr. Ragni and colleagues highlighted how culturing adipose-MSCs (ASCs) in different artificial media able to mimic the inflamed joint environment or osteoarthritis patients’ synovial fluid might drive MSC-secretome towards an increased remodelling capacity of the diseased extracellular matrix and the ability to promote the homeostasis of inflammatory cells, prompting macrophages towards an anti-inflammatory M2 phenotype. Thus, preconditioning in an inflammatory environment can modulate ASCs’ therapeutic capacities and their soluble molecules/EVs for future applications, either cell-based or cell-free, in the field of joint pathologies (O-10).

In the same path, the last presentation of this session was given by Dr. Stefania Bruno (University of Torino, Italy) on the role of purified EVs derived from human liver stem cells (HLSC-EVs) in chronic kidney disease (CKD) and cardiac dysfunction in the remnant kidney murine model [8]. The study provides a foundation for both HLSC contribution to tissue regeneration in liver and kidney injuries and the retained therapeutic properties of HLSC-EVs. The illustrated results proved the capability of HLSC-EVs to ameliorate renal function and reduce interstitial fibrosis, glomerular fibrosis, and capillary rarefaction in an SCID mouse model of chronic kidney disease. This improvement in function was further supported by a reduction in pro-fibrotic and pro-inflammatory gene expression in the renal tissue. Also, EVs reduce cardiac interstitial fibrosis and improve cardiac function, both of which are symptoms of uremic cardiomyopathy and key hallmarks of diastolic dysfunction. Therefore, HLSC-EVs might be a feasible cell-free option to slow down the development of CKD and related uremic cardiomyopathy.

Overall, all presentations showed how cell-free therapies based on single molecules, an array of factors as the secretome, or specialised particles as EVs have the potential to be envisioned as therapeutic and regenerative agents for several pathologies, either used per se or as a scaffold/inspiration for engineered synthetic biology or material science.

### 2.4. Session 4: “Tips and Tricks of Research Valorisation”

The following session on the second day of the congress aimed to discuss how to valorise scientific research and was opened by Dr. Luigi Nicolais from Naples, Italy, who presented the “Materias” business model (O-11). The path from basic research to a commercial product or process is long and complicated. In many cases, both innovators and investors often refer to a “funding gap” that exists between the conception of new ideas and the commercialisation of these innovations. In fact, researchers are not necessarily innovators and, therefore, may not have the necessary skills to transform research results into viable commercial products. For this reason, Horizon Europe has developed a European Innovation Council to promote an entrepreneurial culture and facilitate the dissemination of scientific results, providing support for the commercialisation of research results. Furthermore, the full industrial potential of Italian scientific research remains unrealised, and this deficiency can be traced back to inadequate investments, a fragile venture capital landscape, and a lack of expertise in efficient technology transfer. The “Materias” business model aims to connect academic research and industrial companies in the field of advanced materials and promote new business opportunities by leveraging research results.

An example of a scientific research investment company was presented by Dr. Lucilla Mazzeo from Rome, Italy, who introduced Scientifica Venture Capital, an investment holding company that selects projects and startups with high technological content (O-12). This company was founded in 2021 and provides both financial support and access to the laboratories and equipment necessary to develop the projects. It invests in advanced manufacturing, advanced materials, artificial intelligence, and quantum technologies in the pre-seed, seed, and early-stage sectors and has already funded eight start-ups. The simultaneous ability to support innovative companies in the tech sector and its positioning in the market make Scientifica an innovative company capable of promoting the excellence of Italian research at an international level and underlining its importance as an essential driver of economic competitiveness by basing the technology transfer model on investments, services, and people.

Regarding efficient technological transfer, Dr. Manuela Monti from IRCCS Istituto Romagnolo per lo Studio dei Tumori (IRST) ‘‘Dino Amadori”, Meldola, Italy, spoke about Advanced Therapy Medicinal Products (ATMP), which represents the new frontier for the treatment of numerous rare genetic, onco-haematological, and difficult-to-treat chronic degenerative pathologies. Currently, of the 17 ATMPs authorised by the European Union, 4 are derived from Italian academic research. The aim of basic research must not only be scientific publication but also the development of new therapies which, to reach clinical practice, must take into consideration preclinical research, clinical research, GMP production, feasibility, and sustainability. It is, therefore, essential to establish effective strategies between academic institutions and other stakeholders, with the aim of accelerating and optimising the regulatory process and identifying the main critical issues related to the development of ATMPs (O-13).

Afterwards, Dr. Aubrey Lambert from the company Tomocube, South Korea, introduced the Holotomography (HT) microscope. It is a 4D quantitative imaging solution for the dynamics and mechanisms of live cells, subcellular organelles, and tissue structures without using any sample preparation technique. HT uses the refractive index (RI), an intrinsic optical parameter describing the speed of light passing through a specific material to visualise living cells and tissues. It illuminates the sample with various beam patterns to capture a sequence of holograms from different positions. Most microscopy techniques limit the quantity and quality of information available to researchers, constrain the study of thicker specimens, and even harm the cells during long-term studies. This new technique overcomes these limitations, allowing it to work with confluent cell sheets and significantly reducing laser-induced speckle noise for enhanced contrast. The audience was amazed by the images shown in the presentation, showing how astonishing this new technique is.

The last speaker, Dr. Marco Lorenzi from AlfatestBio, Milan, illustrated the characteristics of BC43, an innovative bench-top confocal microscope. Such machinery is innovative as it is small enough to fit on benches of any kind of biological laboratory, allowing the approach to this technique to anyone interested in implementing imaging in routine work. It is simple to use and does not require specific expertise. Samples are treated with fluorescent dyes and added to the microscope. Images can be viewed with five different lenses and four different excitation wavelengths. The videos shown by Dr. Lorenzi were inspiring, and such instruments are a revolution for many laboratories active in research.

### 2.5. Session 5: “Stem Cells and Cancer”

The afternoon session entitled “Stem cells and cancer” began with a keynote lecture (O-14) by Prof. Ralf Hass from Hannover Medical School, Germany, explaining the role of MSCs and MSC-derived products in a tumorigenic environment and their potential use in translational clinical treatments. Since MSCs represent a cell population with multipotent properties and a heterogeneous composition, they include interdependent types of distinct stroma and stem-like cells rather than a uniform population. A comparison of various tissues revealed that MSCs obtained from birth-associated products represent a more naïve stem cell phenotype with superior regenerative potential [9]. MSCs are activated at damaged tissue sites in a pro-inflammatory environment where they initiate repair processes and regulate tissue homeostasis, but the regenerative potential of MSCs is also displayed in a pro-inflammatory environment of invasively growing tumours. Thus, the regenerative properties and repair activities of MSCs do not distinguish between normal and neoplastic tissue damage. Consequently, MSC activities eventually can also support tumour growth [10].

During repair activities, MSCs perform direct cell-to-cell interactions and paracrine effects on surrounding cells by permanently releasing trophic factors and extracellular vesicles (EVs) including exosomes [11]. Based on the close interactions with cancer cells, MSCs and their derived exosomes can be used to target tumours. Indeed, previous works suggested that paclitaxel-incubated MSCs can have anti-tumorigenic effects on cancer cells [12].

Subsequent ideas were presented by Dr. Giulia Stella from Pavia, Italy, and discussed in this workshop (O-15) for a Phase I clinical trial to test the local delivery of drug-loaded MSCs as a vehicle for local delivery of chemotherapy agents in pleural mesothelioma (PM). The use of paclitaxel-loaded MSCs to treat PM should provide several potential advantages over systemically administered drugs, including reduced toxicity and effective drug concentration at the tumour site. Previous data from Prof. Pessina’s lab. have shown that, in in vivo models, the intra-tumour, as well as the systemic injection (i.v) of MSCs loaded with PTX, strongly inhibit tumour growth [12]. This pharmacological tool is conveyed within the pleura via the endothoracic drainage tube placed at the end of the routine diagnostic thoracoscopy procedure and before pleural talcade [13]. This approach ensures that there are no additional risks associated with the administration of an experimental therapeutic tool. The use of PTX-loaded MSCs to treat PM should provide several potential advantages over systemically administered drugs as follows: (i) it may increase the protection of the drug from degradation before reaching the target cancer cells; (ii) MSCs are capable of integrating into the tumour stroma (MSCs homing capacity), which may enhance the drug concentration into the tumour environment and thus increase tumour drug uptake; (iii) drug delivered by the cells can be better localised in the tumour, thereby reducing its interaction with normal cells and contributing to decreasing systemic toxicity; (iv) a reduced amount of drug could be necessary for treating patients. Based on the results obtained using the procedure established by the methodology patented by the Besta Institute (PCT/EP2011/059626), the new product named Paclimes will be produced using a large-scale methodology in a bioreactor [14]. This new therapeutic approach exploits the natural ability of MSCs to incorporate PTX in vitro and then release it in the area of tumour, thereby contributing to the prevention or reduction of tumour growth or relapse after surgery.

The potential therapeutic use of paclitaxel-incubated MSC was also discussed by Dr. Benedetta Ferrara from IRCCS San Raffaele, Milan, Italy (O-16), for the treatment of pancreatic ductal adenocarcinoma (PDAC), particularly liver metastases derived from it. The complex tumour microenvironment and dense fibrotic stroma in PDACs limit the penetration of drugs, thereby reducing the effectiveness of conventional therapies [15]. Thus, mice with PDAC which had already developed liver metastases were treated with paclitaxel-incubated MSCs. This therapeutic approach not only targeted the primary tumour but also markedly reduced the metastatic burden. Moreover, intraportal injection of paclitaxel-incubated MSCs appeared to be more effective compared to systemic intravenous application, revealing a high and prolonged biodistribution of paclitaxel-incubated MSCs accumulating in the liver.

While paclitaxel-incubated MSCs may provide a cellular system for targeting cancer cells, isolated exosomes derived from paclitaxel-incubated MSCs would represent a cell-free system as a preferable alternative application in some cases.

Thus, Dr. Eleonora Spampinato from the IRCCS Neurologic Institute C. Besta Foundation, Milan, Italy, presented her data for the potential use of EVs/exosomes as a cell-free approach to treat mesothelioma patients. EVs/exosomes from paclitaxel-stimulated adipose tissue-derived MSCs were isolated under GMP conditions and demonstrated stability and product integrity (O-17). Following further tests, this protocol of Dr. Spampinato for vesicle preparation suggests a possible future use in mesothelioma applications.

Further tumour entities were discussed in this workshop by Prof. Hass, including breast cancer, ovarian cancer, lung cancer, astrocytoma, and colon carcinoma with respect to the application of tumour-therapeutic exosomes obtained from drug-loaded MSCs. In more detail, exosomes isolated from taxol-treated MSCs significantly killed cancer cells in vitro and markedly reduced tumours, particularly distant organ metastases in vivo [16].

In summary, these findings and workshop discussions underscored the therapeutic potential of drugged MSCs and/or derived exosomes as vehicles to deliver anti-tumor compounds, as displayed in Figure 1.

Conventional therapies with a non-specific systemic drug application are limited and have severe side effects. In contrast, the advantages of using immunologically inert and drugged MSCs or derived cell-free exosomes are obvious since these vehicles can specifically target primary tumours and, most importantly, tumour metastases. Consequently, drugged MSCs and/or derived exosomes provide a promising translational platform for the potential entry of this anti-tumour strategy into the clinic.

### 2.6. Session 6: “Stem Cells in Veterinary Applications”

The veterinary session of the StemNet meeting was chaired by Dr. Silvia Dotti (IZSLER, the Zooprophylactic Institute of Lombardy, and Emilia-Romagna, Brescia, Italy) and Dr. Anna Lange Consiglio (University of Milan, Italy) as members of the GISMvet (Veterinary Section of Italian Mesenchymal Stem Group). GISMvet was created in 2021 by a group of veterinary doctors with many years of experience in regenerative medicine, with the aim of establishing a connection between the scientific community and veterinary practitioners, sharing biological knowledge of MSC-based therapies and novel strategies and challenges for their clinical applications. In the “One Health-One Medicine” approach, veterinary clinical trials and research represent an important contribution to human studies, as similarities between spontaneous disease in domestic animals can be found with their human counterparts.

The first invited speaker of the session, Prof. Steven Dow from Colorado State University, USA, described the extensive work developed under his guidance on the use of MSCs for the management of drug-resistant bacterial infections [17]. Antibiotic resistance is a global public health concern, which should be addressed through the “One Health” approach. Firstly, Prof. Dow illustrated the therapeutic rationale for using MSCs for this purpose, based on their studies on in vitro modelling of biofilm disruption activity (O-18). The relevant mechanisms of action include the intrinsic ability of MSCs to produce and secrete multiple antimicrobial peptides which are effective against multiple species of bacteria. In the second part of his presentation, prof. Dow shared results from the following in vivo and clinical studies: (1) mouse Staphylococcus aureus biofilm injection model, (2) canine patients with spontaneous drug-resistant wound infections, and (3) equine MRSA septic arthritis models. In all studies, activated allogeneic adipose-derived MSCs were administered either intravenously (study 1 and 2) or intra-articularly (study 3). The results demonstrated that MSC-based therapies are efficient in promoting the microbiological clearance and clinical resolution of infections. From a clinical perspective, Prof. Dow reported that MSCs need to be administered multiple times for the treatment of chronic infections and that they have a synergistic and additive effect with all major classes of clinically relevant antibiotics.

Dr. Ana Ivanovska, from the University of Galway, Ireland, was the second invited speaker, and she presented the activities of the CALIN (Celtic Advanced Science Innovation Network) Veterinary Regenerative Network [18]. The initiative recognised the need to share and integrate knowledge from human and animal cell therapies as essential for the advancement of the field of regenerative medicine under the “One Health–One Medicine” umbrella. Preliminary results (O-19) from a survey showed that the current major hurdles for the wide application of cell therapies in the veterinary clinical practice are costs, availability, and efficacy. In terms of manufacturing standards, a position statement publication highlighted the need to design a disease-specific potency assay that can efficiently correlate in vitro findings with desired clinical outcomes. In addition, the species-specificity of growth factors has been identified as an important factor for the expansion of MSCs for therapeutic purposes. A major current gap is the lack of consolidated veterinary regenerative medicine courses in veterinary schools worldwide. To address this need, eight free continuous professional development (CPD) CALIN webinars on veterinary regenerative medicine were delivered in collaboration with professionals from European and American veterinary schools.

Finally, Dr. Priscilla Berni from the University of Parma, Italy, ended the session with a selected oral communication (O-20) about one of the main challenges occurring when moving from the bench to the bedside: the shipment conditions of MSCs. Time, temperature, and suspension products are the three key aspects. Based on these, fresh cultured cells are suitable for short-term storage, and the suspension vehicle can be directly administered to the patient. For long-term shipment, cryopreserved cells seem to be a better option, but manipulations are required to remove xenobiotic/cryoprotectant products from the medium before in vivo use. In this study, both fresh and cryopreserved cells derived from canine adipose tissue were stored in nutrient-poor and nutrient-rich vehicles at room temperature for the short and long term. Cells were compared in terms of mortality, metabolic activity, and replicative capacity. Furthermore, the expression analysis of a panel of genes involved in MSCs’ biological features was investigated. According to the literature, no significant differences were found between fresh and frozen cells, and the results suggested that MSCs could be maintained between 2 and 4 h in both vehicles, but resuspension in poor vehicles does not ensure cell viability for up to 24 h. Different vehicles can also modify the expression of the genes involved in the immunomodulatory activity of MSCs. Future evaluations using different volumes and concentrations to stimulate the doses used for clinical applications should be investigated.

### 2.7. Session 7: “Stem Cells in Clinical Applications”

Stem cells are classified as advanced therapy medicinal products (ATMP) that fall under drug rules. For this reason, the translation of these new therapeutic products from the laboratory to the market must be conducted under highly defined regulations and directives provided by competent regulatory authorities. Hematopoietic Stem Cells (HSCs) and MSCs hold significant potential for various clinical applications, including tissue regeneration, disease treatment, and organ repair, but are also promising in various clinical applications, offering potential therapeutic effects for regenerating damaged cells and assisting in organ recovery [19]. In Europe, the main applicable laws in this field are as follows: Regulation (EC) No. 1394/2007 [20], which includes the requirements to be used in the development, manufacturing, or administration of ATMPs’ Directive 2001/83/EC, which applies to medicinal products for human use; and Regulation (EU) No. 536/2014 [21], which applies to clinical trials on medical products intended for human use. After presenting an application to the regulatory authority responsible for clinical trial oversight (FDA or EMA), the application will be reviewed in accordance with the FDA/EMA criteria and, if assured of the protection of humans enrolled in the clinical study, the application will be approved by the investigational review boards (IRBs) in the United States or Ethics Committees (ECs) in the European Union.

Dr. Diego Ponzin (Fondazione Banca degli Occhi del Veneto, Venice, Italy) spoke about the importance of human corneal endothelial cells (HCECs), which can be cultivated in vitro to create a sheet of stem cells that can be used instead of actual corneas during transplants (O-21). The major complication is that these cells do not normally replicate in vivo but innovative protocols are being designed for use in normal clinical practice. An example of the application of these cells is age-related macular degeneration, which results in photoreceptor degeneration. By using cell-based therapies, such as the implementation of HCECs, researchers have proven that photoreceptors can be rescued and macular degeneration can be prevented in preclinical models.

The session then proceeded with Dr. Tobias Winkler’s (Center for Musculoskeletal Surgery, BIH Center for Regenerative Therapies, Julius Wolff Institute Charité-Universitaetsmedizin Berlin) speech on a new phase III study in which MSCs derived from the placenta are used to treat muscle injuries and surgery-related stress (O-22). Based on his results, Dr. Winkler described a decrease in immunological stress after surgery with the implication of MSCs, measured through the evaluation of biomarkers. This led the group to design the HIPGEN phase III trial. From these data, it was easily appreciated how MSCs can also be used for the treatment of skeletal muscle regeneration, suggesting these types of cells as an innovative therapy in the orthopaedic field.

Subsequently, there was a presentation by Dr. Conti (Department of Veterinary Science, University of Parma, Parma, Italy) on the effect of stromal cells in musculoskeletal (MSK) disorders. The stromal vascular fraction (SVF) consists of a heterogeneous population of stem cells obtained from adipose tissue by enzymatic digestion. By obtaining SVF from adipose tissue through a mechanical approach without substantial manipulations, Dr. Conti’s research group was able to perform micrografts in in vivo procedures. Moreover, the SVF cells were able to proliferate and differentiate in vitro into adipocytes, osteocytes, and chondrocytes. For these reasons, the application of such cells in vivo demonstrated rapid healing in MSK disorders, suggesting an ulterior alternative and application of a type of stromal cell.

In conclusion, Dr. Nicholas Crippa Orlandi (Department of Orthopaedics and Traumatology, University of Siena, Siena) discussed the state of the art in regenerative orthopaedic medicine with the precious insight of someone who has experienced the application of such techniques in orthopaedic surgery (O-23). He proceeded to discuss how the implementation of new strategies can increase the osteogenic properties of MSCs derived from bone marrow, especially those cultured on specific scaffolds. An example of a new strategy is the use of platelet-rich plasma (PRP), which can be substituted with foetal bovine serum (FBS). By utilising this supplement, these scaffolds are a more valid tool for surgeons in difficult orthopaedic surgery conditions.

### 2.8. Session 8: “Organoids and 3D Systems”

An inspiring session titled ‘Organoids and 3D Systems’ was organised, which allowed us to focus on the use of these innovative techniques as they are slowly being considered to replace in vivo experiments. Dr. Molly Stevens, London, UK, spoke about the recent remarkable advancements in basic and translational research, underscoring the pivotal role of organoids, 3D matrix-based cellular systems, and organ-on-chip technologies in different research domains. The innovative approaches presented are poised to shape the future of biomedical research, contributing to breakthroughs in drug development, disease modelling, and regenerative medicine. The ability to regenerate damaged tissues is one of the great challenges in the fields of tissue engineering and regenerative medicine. Dr. Stevens’s group is developing approaches to control cell behaviour through their innate ability to sense and respond to local meso-, micro-, and nanoscale patterns of chemistry, stiffness, and topography. Polymer systems can be functionalised for drug delivery and with biological and synthetic cues to instruct the entire lifecycle of the tissue from cell binding and differentiation to cell-induced material remodelling and ultimate tissue organisation and function. These materials can also be used as platform systems to study a wide variety of instructive environments for tissue regeneration and cell fate due to the adoption of cutting-edge material analysis approaches, such as live cell spectroscopy and correlative nanoscale resolution imaging approaches.

Next, Dr. Silvia Scaglione, CNR Genoa, Italy, spoke about the generation of Multi In Vitro Organ (MIVO) microphysiological systems (MPS) as a new platform to cultivate 3D cancer tissues under physiologically relevant conditions (O-24). The data obtained were compared with the in vivo results, and the comparison validated the accuracy of the 3D model. The presented data confirmed the successful generation of relevant human disease models using MIVO MPS systems.

The session proceeded with Dr. Martyna Malgorzata Rydzyk from Bologna, who spoke about the generation of spheroids made of two different types of osteosarcoma (OS) cells and tumour-associated cells (TAC) (O-25). Thus, it would be an efficient way to mimic and represent in vitro the in vivo tumour behaviour. Dr. Rydzyk illustrated the differences in OS cells cultured alone or in the presence of TACs; in the second setting, the spheroids grew more compact and organised, but most importantly, more viable. Further studies have to be conducted, and these structures are implemented in the understanding of tumours.

To conclude the session, Dr. Francesca Paris from the University of Bologna showed how stem cells can be useful in the understanding of complex diseases such as type 1 diabetes mellitus (O-26). By using perinatal-derived stem cells in the form of spheroids, the research group was able to study the microenvironment of such complex metabolic diseases more accurately. The aim was to recreate a reliable 3D model. They co-cultured spheroids derived from amniotic epithelial cells (AEC) and Wharton’s jelly mesenchymal stromal cells (WJ-MSC). This co-culture was stable, viable, and produced an extracellular matrix. It was also seen that co-cultured spheroids promoted a tolerogenic response by decreasing immune cell proliferation compared to the PBMC control. The analysis of the immunomodulatory and differentiative capabilities of such models is of great interest, especially in the optics of a new cellular therapy solution for type 1 diabetes mellitus.

### 2.9. Session 9: “iPSC and Gene Therapy”

The last session of the congress was opened by Dr. Alessandro Aiuti from San Raffaele of Milan, who introduced how gene therapy based on stem cells and specifically induced pluripotent stem cells (iPSCs) is rapidly becoming an important field of study to provide in vitro models for the study of complex pathologies in order to study the pathological mechanisms underlying diseases and propose new therapeutic treatments.

Indeed, Dr. Claudia Compagnucci from Bambino Gesù Children’s Hospital, IRCCS, Rome, presented the iPSCs as an alternative in vitro model of neurological diseases compared to the in vivo model (O-27). Human physiology is unique, and sometimes the animal model does not recapitulate the human phenotype. The iPSCs are derived from adult somatic cells (i.e., skin fibroblasts) that are genetically reprogrammed into a stem cell-like state. Furthermore, iPSCs can be successfully differentiated into somatic cells, such as neurons and astrocytes, which are useful in the study of neurodegenerative and neurodevelopmental diseases such as riboflavin transporter deficiency (RTD). To understand the organic mechanisms altered in RTD, iPSCs have been used, and this in vitro model allows us to recapitulate and reproduce the initial pathology in a patient-specific manner.

Subsequently, Dr. Elena Laura Mazzoldi from the University of Brescia, Italy, spoke about iPSC as an unlimited and non-invasive source for the isolation and expansion of hyalocytes (O-28). Hyalocytes are a small population of macrophage-like cells that reside in the vitreous cortex of the eye and are involved in physiological and pathological processes at the vitreoretinal interface. Over the years, hyalocytes have been isolated from animals, but there are few studies on cultured human hyalocytes since their isolation requires invasive surgery. In the study presented by Dr. Mazzoldi, the iPSCs were grown on multiwell plates coated with Matrigel, initially differentiated into hematopoietic stem cells (HSCs) using a commercial kit, and then into macrophages using macrophage colony-stimulating factor (M-CSF) treatment. The macrophages were subsequently left untreated (NT) or treated with ascorbic acid alone or in combination with basic fibroblast growth factor (bFGF) and/or transforming growth factor beta 1 (TGFβ1). As a positive control, macrophages were also cultured in the presence of a pool of vitreous bodies obtained from vitrectomies. Cells were analysed morphologically and for gene and protein expression via qRT-PCR, Western blotting, immunofluorescence, and flow cytometry. Similar to vitreous-treated cells, ascorbic-treated macrophages with acid alone or in combination with bFGF presented a more elongated shape compared to NT or cells treated with TGFβ1 so the hyalocytes can be differentiated by treating iPSC-derived macrophages with ascorbic acid alone.

The session was concluded by Dr. Giulia Sofia Marcotto, who used iPSCs to generate a model of multiple system atrophy with predominant Parkinsonism (MSA-P) to characterise it morphologically and functionally (O-29). MSA-P is a sporadic, adult-onset, fatal neurodegenerative disease with a severe clinical outcome, affecting various neuronal pathways including the mesencephalic district of the nigrostriatal system. The iPSC lines were generated by reprogramming peripheral blood mononuclear cells from MSA-P and healthy donors and were differentiated into floorplate (FP)-derived midbrain dopaminergic (DA) neurons. Neurons positive for tyrosine hydroxylase (TH) and with the DA phenotype were evaluated morphologically and functionally using microelectrode systems (MEA). The DA neurons showed a significant reduction in dendritic length and other areas, and MEA recordings showed less stability, typical of an immature cell, revealing both a different morphology and functional phenotype in neurons in MSA-P iPSC-derived mesencephalic cultures.

## 3. Conclusions Remarks

Over the last few years, this congress has had a strong interdisciplinary and innovative perspective, open to planning specific sessions for the most innovative and breakthrough approaches related to stem cells and neighbouring research topics. The conference served as a significant forum for discussion and exchange of ideas, allowing participants to deepen their understanding of relevant topics in the field of MSC implementation. Plenary sessions and poster presentations provided a comprehensive overview of challenges, innovations, and future perspectives. The interaction among experts, researchers, and professionals facilitated the creation of meaningful connections and collaboration on future projects that will be seen in the coming months and years. Moreover, we have seen the active participation of a large number of young researchers who enthusiastically brought ideas and new ways of doing and communicating science.

It is hoped that the insights and reflections emerging from the conference will contribute tangibly to the advancement of the field and the realisation of concrete solutions to current and future challenges.

## 4. Poster Award

During the 2023 StemNet meeting, three “Young Investigator Awards” of 500 euros each were assigned. In order to be eligible, researchers had to (a) submit a spontaneous candidature; (b) be the first author of an accepted abstract; (c) be younger than 35 years on 20 October 2022; (d) be present at the Award Ceremony held during the “GISM—Next Generation” section. The three winners of the 2023 edition were Bari Elia, Cadelano Francesca, and Orlandi Giulia.

## Figures and Tables

**Figure 1 ijms-25-02221-f001:**
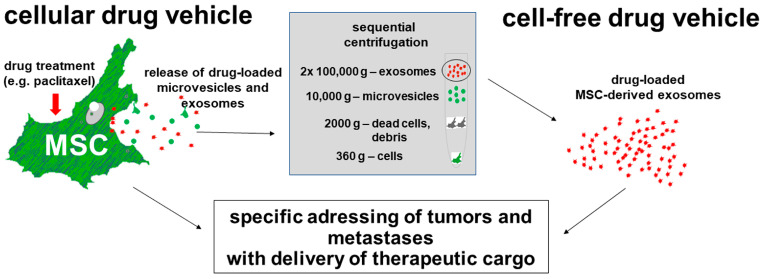
Description of the therapeutic potential of drugged MSCs as cellular vehicles and/or derived exosomes as cell-free vehicles to deliver anti-tumor compounds.

## Data Availability

The research data associated with the presentations reported in this manuscript are reported in the Appendix A.

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
