# Peer review of "State of the Art and New Trends from the Second International StemNet Meeting"

_ijms, 2024, doi:10.3390/ijms25042221_

Round 1
Reviewer 1 Report
Comments and Suggestions for Authors
I found the conference report on the second international Stemnet meeting to be quite engaging. The authors have done an excellent job summarizing the diverse contributions presented during the congress. Overall, the work is both clear and comprehensive, although some contributions appear to be somewhat succinctly summarized.
I have only a few minor comments regarding this work. In Session 6, titled "STEM CELLS IN VETERINARY APPLICATIONS," the authors explicitly mention the affiliations (for instance, "The veterinary session of the StemNet meeting was chaired by Dr. Silvia Dotti from IZSLER, the Zooprophylactic Institute of Lombardy and Emilia-Romagna, Brescia, Italy, and Dr. Anna Lange Consiglio from the University of Milan, Italy"). However, in the subsequent Session 7, titled "STEM CELLS IN CLINICAL APPLICATIONS," the author's affiliations are no longer described. This inconsistency is observed in other chapters as well. I would recommend harmonizing this aspect throughout the entire text.
Additionally, the chapter labeled "3. Conclusion remarks" may benefit from being rewritten and expanded. It should offer a more extensive commentary on the current topics discussed during the congress and provide insights into future directions.
Author Response
Dear reviewer,
please find enclosed the revised manuscript entitled “STATE OF THE ART AND NEW TRENDS FROM THE SECOND INTERNATIONAL STEMNET MEETING”
Thank you for the comments. We have proceeded to correct and make more coherent the affiliations of the authors.
We also provided to lengthen the conclusions and make them more conclusive on the matter.
All alterations in the revised manuscript are tracked using the "Track Changes" function in Microsoft Word.
We believe that following the reviewers’ suggestions the new version has been much improved, and hope that it is now suitable for publication in your journal.
We hope that after the correction the paper will result more coherent and suitable for publication. I remain available for further information and questions.
Yours sincerely
Katia Mareschi
Reviewer 2 Report
Comments and Suggestions for Authors
Excellent work. Total number of attendant in this conference might be written in Introduction. Copyright of Fig. 1 is OK?
Author Response
Dear reviewer,
please find enclosed the revised manuscript entitled “STATE OF THE ART AND NEW TRENDS FROM THE SECOND INTERNATIONAL STEMNET MEETING”
Thank you for the comments. We have inserted in the introduction the total number of participants who have attended the congress, both presenters and attendees.
The copyright of Fig. 1 is ok as the figure has been created by Prof. Hass ad hoc for the article and hasn’t been published in other papers before.
All alterations in the revised manuscript are tracked using the "Track Changes" function in Microsoft Word.
We believe that following the reviewers’ suggestions the new version has been much improved, and hope that it is now suitable for publication in your journal.
Kind regards,
Katia Mareschi